# The Natural Compound CalebinA Suppresses Gemcitabine Resistance and Tumor Progression by Inhibiting Angiogenesis and Invasion Through NF-κB Signaling in Pancreatic Cancer

**DOI:** 10.3390/nu17162641

**Published:** 2025-08-14

**Authors:** Yuki Eguchi, Yoichi Matsuo, Masaki Ishida, Yuriko Uehara, Saburo Sugita, Yuki Denda, Keisuke Nonoyama, Hiromichi Murase, Tomokatsu Kato, Kenta Saito, Takafumi Sato, Hiroyuki Sagawa, Yushi Yamakawa, Ryo Ogawa, Hiroki Takahashi, Akira Mitsui, Shuji Takiguchi

**Affiliations:** 1Department of Gastroenterological Surgery, Graduate School of Medical Sciences, Nagoya City University, 1-Kawasumi, Mizuho-cho, Mizuho-ku, Nagoya 467-8601, Japan; yukieg0802@gmail.com (Y.E.); i.masaki0103@gmail.com (M.I.); k.yuriko31@gmail.com (Y.U.); ssabu3753@gmail.com (S.S.); denda@med.nagoya-cu.ac.jp (Y.D.); knonoyama0924@gmail.com (K.N.); muramen5.com@gmail.com (H.M.); tomo.k.g.w@gmail.com (T.K.); kentaxis777@gmail.com (K.S.); tsato.ncu@gmail.com (T.S.); hiro18.hiroyuki.hero@gmail.com (H.S.); uc19810116@gmail.com (Y.Y.); ryogawancu@gmail.com (R.O.); coolsound1230@gmail.com (H.T.); a.mitsui.21@west-med.jp (A.M.); takiguch@med.nagoya-cu.ac.jp (S.T.); 2Department of Gastroenterological Surgery, East Medical Center, Graduate School of Medical Sciences, Nagoya City University, Nagoya 467-8601, Japan

**Keywords:** CalebinA, gemcitabine resistance, NF-κB, pancreatic cancer, natural compound, angiogenesis, invasion

## Abstract

**Background:** Previously, we established gemcitabine (Gem)-resistant pancreatic cancer (PaCa) cell lines and showed that the acquisition of Gem resistance is accompanied by enhanced activation of the inflammatory transcription factor nuclear factor-κB (NF-κB). In this study, we focus on CalebinA, a natural compound derived from the rhizomes of turmeric, known for its potent anti-inflammatory properties. It has been suggested that this compound may exert anticancer effects by downregulating the NF-κB signaling cascade. Therefore, we collaborated with Sabinsa Corporation, Japan, to explore its potential application in pancreatic cancer therapy. **Methods:** We used gemcitabine-resistant pancreatic cell lines to demonstrate the effect of CalebinA on cell toxicity, invasiveness, cytokine levels, NF-κB p65 activity, and tube formation in angiogenesis. Tumor volume and histopathological analysis were used to analyze the effects of CalebinA on tumors induced by the subcutaneous injection of pancreatic cell lines in mice. **Results:** Treatment with 10 μM CalebinA significantly inhibited NF-κB activity. Gem-resistant PaCa cells exhibited higher invasive and angiogenic capacities than non-resistant parental cells; however, these capacities were markedly suppressed by CalebinA. In vivo, intraperitoneal CalebinA administration every 3 days led to a significant reduction in tumor volume in mice bearing subcutaneous xenografts of the AsPC-1 pancreatic cancer cell line. Immunohistochemical analysis revealed that CalebinA suppressed the expression of Ki-67, CD31-positive microvessel density, and NF-κB p65. **Conclusions:** These findings suggest that CalebinA holds promise as a novel therapeutic agent for Gem-resistant pancreatic cancer and may be a strong candidate for clinical application.

## 1. Introduction

Pancreatic cancer is one of the most lethal malignancies among all cancers. In 2023, it ranked as the third leading cause of cancer-related deaths in both Japan and the United States [1,2]. The prognoses remain extremely poor, with a reported 5-year survival rate of less than 10% [3,4]. With increasing mortality rates among patients with pancreatic cancer, it is projected to become the second leading cause of cancer-related deaths in the United States by 2030 [5]. Similarly, in Japan, both the occurrence of new cases and cancer-related mortality have been steadily increasing in pancreatic cancer since the 1960s [6].

Numerous studies have been conducted on chemotherapy for pancreatic cancer, and various attempts have been made to develop novel therapeutic agents [7,8,9]; however, compared to other types of cancer, treatment options for pancreatic cancer remain limited. Its aggressive nature, characterized by early dissemination and treatment refractoriness, contributes to the poor prognosis [10,11]. The development of new therapeutic strategies for pancreatic cancer is an urgent priority.

As part of efforts to improve treatment outcomes, a large-scale phase III trial was initiated in 1997 targeting patients with pancreatic cancer to compare gemcitabine monotherapy with fluorouracil monotherapy, which was the standard treatment at the time. Gemcitabine was found to extend survival more effectively than fluorouracil, with median durations of 5.7 and 4.4 months, respectively [12]. For cases of unresectable or metastatic pancreatic cancer, gemcitabine became the standard first-line chemotherapy following its approval, and has maintained its position as the preferred therapeutic approach in clinical practice to this day [13]. In 2013, the Prep-02/JSAP05 study was undertaken to investigate whether preoperative chemotherapy with gemcitabine and S-1 could provide a survival benefit over immediate surgery. Findings showed a considerable increase in median overall survival in the pretreated group (36.7 months) as opposed to 26.6 months in those treated with surgery alone [14]. However, despite the optimistic findings of the trial, clinical data accumulated suggest that the therapeutic effect of gemcitabine has only a transient effect on pancreatic cancer in the initial phase, with its efficacy diminishing rapidly. Thus, the development of gemcitabine resistance remains a major clinical hurdle. However, the underlying mechanisms of this resistance have yet to be fully elucidated; clarifying the mechanisms of gemcitabine resistance is expected to lead to improved therapeutic strategies for pancreatic cancer.

To investigate the mechanisms underlying gemcitabine resistance, we successfully established gemcitabine-resistant pancreatic cancer cell lines. Evidence was obtained suggesting that the acquisition of gemcitabine resistance enhances nuclear factor κB (NF-κB) activity [11,15,16]. The transcription factor NF-κB is the key to controlling various cellular processes, including proliferation, programmed cell death, and inflammation. It also contributes to cancer progression by promoting invasion, metastasis, and other oncogenic pathways [17,18]. Furthermore, pro-inflammatory cytokines such as IL-1 and TNF-α are not only target genes of NF-κB, but also potent activators of NF-κB signaling. In the context of pancreatic cancer, NF-κB signaling has emerged as a key modulator of angiogenesis, which is a process indispensable for neoplastic growth and metastatic dissemination [19,20]. Moreover, many chemotherapeutic agents, including gemcitabine, have been reported to activate NF-κB [21]; inhibiting NF-κB alongside chemotherapy holds promise as a new therapeutic strategy against pancreatic cancer.

Numerous natural compounds are recognized for their favorable safety profiles and have been extensively studied for their anti-inflammatory and anticancer mechanisms [22,23]. The rhizome of turmeric is a widely used traditional plant-derived product in India and China. Turmeric has been shown to possess a wide range of biological properties, including anticancer, anti-inflammatory, antioxidant, anti-arthritic, anti-aging, antimicrobial, and wound-healing effects [24,25,26]. Most research on turmeric has primarily focused on its active compound, curcumin [27]. However, CalebinA, another bioactive compound found in turmeric rhizomes, has been receiving increasing attention recently. CalebinA is an active constituent isolated from turmeric extracts that do not contain curcumin, and it exerts antineoplastic effects, in part, by disrupting multiple oncogenic signaling pathways, with NF-κB being one of the most affected [28,29,30]. Therefore, in this study, we collaborated with Sabinsa Corporation, Japan, which successfully isolated and produced CalebinA, with the objective of exploring its potential application in the treatment of pancreatic cancer.

We first investigated whether low-dose CalebinA suppresses NF-κB activity. Following this, we investigated its role in regulating angiogenesis in pancreatic cancer cells. We found that CalebinA inhibits pancreatic cancer angiogenesis both in vitro and in vivo by reducing NF-κB activity and the associated production of angiogenic factors.

To our understanding, this is the first report to demonstrate that CalebinA inhibits NF-κB signaling in pancreatic cancer cell lines and subsequently affects angiogenesis. Furthermore, our findings suggest that CalebinA may serve as a promising therapeutic agent for the treatment of pancreatic cancer.

## 2. Materials and Methods

### 2.1. Reagents

CalebinA was kindly provided by Sabinsa Corporation (Tokyo 171-0022, Japan). A 100 mM stock solution of CalebinA was prepared in DMSO (Sigma-Aldrich, St. Louis, MO, USA; Cat. No. 472301), aliquoted, and stored at −20 °C. Recombinant human tumor necrosis factor-α (TNF-α), a neutralizing monoclonal anti-human VEGF antibody, and an IL-8 antibody were purchased from R&D Systems (Minneapolis, MN, USA; Cat. No. MAB210 MA251N-100 MAB208). Antibodies used in this study included rabbit monoclonal antibodies targeting Ki67 and NF-κB p65, and a rabbit polyclonal antibody against CD31, all of which were purchased from Abcam (Cambridge, UK; catalog numbers: ab16667, ab32536, ab32457).

### 2.2. Cell Lines and Cell Culture

The human pancreatic cancer cell lines AsPC-1 and PANC-1 were purchased from the American Type Culture Collection (ATCC, Manassas, VA, USA; Cat. No. CRL-1682 CRL-1469). The AsPC-1 pancreatic cancer cell line was maintained in RPMI-1640 medium (Roswell Park Memorial Institute formulation; Sigma-Aldrich, St. Louis, MO, USA catalog number R8758), and PANC-1 cells were maintained in Dulbecco’s Modified Eagle Medium (DMEM; Sigma-Aldrich, St. Louis, MO, USA Cat. No. D6429), as previously described [9,11,15,31,32]. Both media were supplemented with 10% fetal bovine serum (FBS), 10,000 U/mL penicillin, 10 mg/mL streptomycin, and 25 µg/mL amphotericin B (all from Gibco, Grand Island, CA, USA; Cat. No. 26140079, 15140122, 15290018).

### 2.3. Development of Gemcitabine-Resistant Pancreatic Cancer Cell Lines

In our earlier research [15], we independently developed pancreatic cancer cell lines with acquired resistance to gemcitabine. The drug was obtained from Eli Lilly Japan K.K. (Kanagawa, Japan; YJ code: 4224403D1030). To evaluate drug sensitivity, IC_50_ values were calculated for each cell line by constructing dose–response curves using the WST-1 cell proliferation assay (Takara Bio, Yamanashi, Japan; Cat. No. MK400). During serial passaging, each cell line was cultured in a medium containing gemcitabine at a concentration equivalent to its IC_50_ for 3–4 weeks. The IC_50_ was then re-evaluated, and the gemcitabine concentration was adjusted accordingly. This process was repeated multiple times. Gemcitabine-resistant cell lines were defined as similar to those with an IC_50_ > 50-fold higher than that of the parental cell line; this resistance was experimentally confirmed.

### 2.4. Cell Viability Assay

The cytotoxicity of CalebinA was evaluated using the WST-1 assay (Takara Bio, Yamanashi, Japan; Cat. No. MK400) according to the manufacturer’s protocol. Gemcitabine-sensitive (Gem-S) and gemcitabine-resistant (Gem-R) AsPC-1 and PANC-1 cells were seeded at a density of 3 × 10^3^ cells per well in a total volume of 100 µL in 96-well plates and incubated for 24 h. Cells were then treated with various concentrations of CalebinA (0–100 µM). After 48 h of incubation, absorbance was measured at 450 nm using a SpectraMax 340 microplate reader (Molecular Devices, Sunnyvale, CA, USA).

### 2.5. Nuclear Fraction Preparation and NF-κB p65 Activity Measurement

NF-κB activity was measured using 10 µg of nuclear extracts with the TransAM NF-κB p65 kit (Cat. No. 40096; Active Motif, Inc., Carlsbad, CA, USA), according to the manufacturer’s protocol. Cells were pretreated with CalebinA (10 µM) for 24 h and with TNF-α (0.3 ng/mL) for 15 min. Nuclear extracts were prepared using the Nuclear Extraction Kit (Cat. No. 40010; Active Motif, Inc. Carlsbad, CA, USA) in accordance with the manufacturer’s instructions. Protein concentration was determined using the BCA Protein Assay Kit (Cat. No. 23225; Thermo Fisher Scientific, Waltham, MA, USA).

### 2.6. Matrigel Invasion Assay

To evaluate the invasive potential of pancreatic cancer cells, we utilized BioCoat Matrigel Invasion Chambers (Corning, NY, USA; Cat. No. 354480), incorporating minor protocol adjustments from earlier publications [15,33]. Cells were seeded into the upper chambers at a density of 1 × 10^5^ cells per chamber, in appropriate serum-free medium. TNF-α (0.3 ng/mL) and a concentration of 10 µM CalebinA was introduced into both the upper and lower compartments of the invasion chamber. To promote chemotactic migration, 10% fetal bovine serum (FBS; Gibco, Grand Island, CA, USA) was placed in the lower chamber. Following incubation, migrated cells were stained using the Diff-Quick staining kit (Sysmex, Kobe, Japan; Cat. No. ZS1009). The number of invaded cells was quantified by counting nine randomly chosen fields under 40× magnification, and the mean cell count was then determined.

### 2.7. ELISA

AsPC-1 (Gem-S and Gem-R) and PANC-1 (Gem-S and Gem-R) cells were seeded in 6-well plates at a density of 1 × 10^5^ cells per well and cultured overnight at 37 °C in an appropriate medium supplemented with 10% FBS. The next day, the medium was replaced with serum-free medium, and cells were treated with or without CalebinA (10 µM) and TNF-α (0.3 ng/mL), followed by incubation for 48 h. Following a protocol from a previous study [9], the concentrations of VEGF and IL-8 in the culture supernatant were measured using ELISA kits (Cat. No. D8000C and DVE00; R&D Systems, Minneapolis, MN, USA).

### 2.8. Tube Formation Assay for Angiogenesis

To evaluate angiogenic potential, a tube formation assay was performed using EA.hy926 cells and the Matrigel Matrix (Cat. No. 354230; Corning, Inc., New York, NY, USA) in accordance with previously described methods [32,33]. AsPC-1 (Gem-S and Gem-R) and PANC-1 (Gem-S and Gem-R) cells were seeded at a density of 1 × 10^5^ cells per well in 6-well plates containing an appropriate medium supplemented with 10% FBS, and incubated overnight at 37 °C. On the following day, the medium was replaced with a medium containing 2% FBS, and the cells were further incubated for 48 h in the presence or absence of CalebinA (10 µM) and TNF-α (0.3 ng/mL). After incubation, the culture supernatant was collected, centrifuged, and used as a conditioned medium. Equal volumes of conditioned medium and culture medium were mixed and added to Matrigel-coated wells along with EA.hy926 cells (1.2 × 10^4^ cells/well), followed by incubation for 12 h. Using the BZ-X710 fluorescence microscope (Keyence Corporation, Osaka, Japan), we examined the formation of capillary-like networks. Intersections were quantified across multiple microscopic fields, and the average value was calculated.

In all in vitro experiments, at least three independent biological replicates were performed, and each experiment included four technical replicates per condition.

### 2.9. Animals

Twenty-four female BALB/c nude (nu/nu) mice (5 weeks old, 14–16 g) were obtained from Charles River Laboratories (Wilmington, MA, USA). The animals were kept in groups of three in a standard Plexiglas cage under controlled environmental conditions (temperature: 20–26 °C; humidity: 40–60%) with a 12 h light/dark cycle. Sterilized chow and water were provided ad libitum. Isoflurane (FUJIFILM Wako Pure Chemical Corporation, Tokyo, Japan) was used for anesthesia, with concentrations of 2.5% for induction and 2.0% for maintenance via inhalation. After confirming sufficient anesthesia, euthanasia was performed through cervical dislocation. All procedures involving animals were approved by the Animal Care and Use Committee of Nagoya City University Graduate School of Medical Sciences (Approval No. Medical Animal 24-027, approved on 1 November 2024). The approval was extended on 24 April 2025, under Approval No. Medical Animal 24-027R24, and is valid until 31 March 2026.

### 2.10. Experimental Protocol

AsPC-1 (Gem-S and Gem-R) pancreatic cancer cells (1 × 10^6^ cells suspended in 100 µL of Hank’s balanced salt solution, Sigma-Aldrich (Merck), St. Louis, MO, USA Cat. No. H9394) were subcutaneously injected into the right flank of each mouse (Gem-S: 12 mice and Gem-R: 12 mice). Tumor volume was measured every 3 days using calipers. We randomly allocated the mice into two groups (n = 6 per group) after their tumors grew to an average volume of approximately 30 mm^3^. Starting on that day, the mice in Group I received an intraperitoneal injection of 400 µL of a control solution containing the same concentration of DMSO and 5% Tween-80 (Sigma-Aldrich, St. Louis, MO, USA) diluted in Milli-Q water, every 3 days. The mice in Group II received an intraperitoneal injection of 400 µL of a solution containing CalebinA (40 mg/kg) and 5% Tween-80 diluted in Milli-Q water every 3 days. This treatment continued for 3 weeks, for a total of seven injections, followed by euthanasia 3 days after the final administration. We would like to explain the rationale for selecting this protocol. To the best of our knowledge, this is the first study to evaluate the in vivo antitumor effects of CalebinA against pancreatic cancer, and there are very few reports on its intraperitoneal administration. While previous studies employed oral administration, we opted for intraperitoneal injections to ensure more accurate and reliable delivery of the compound. The oral dose reported in prior studies was converted to an equivalent intraperitoneal dose, and 40 mg/kg was selected as the maximum concentration at which CalebinA could be solubilized. Regarding the treatment duration, we found that a 4-week administration period led to ulcer formation in the control mice. Therefore, we adopted a protocol of administering 40 mg/kg of CalebinA every three days for a total of three weeks.

The volume of each tumor was estimated using the following formula: (length × width^2^)/2. Here, the length (A) denotes the longest diameter, and the width (B) denotes the shortest diameter, measured in millimeters. After measurement, tumors were surgically removed and immersed in 10% neutral-buffered formalin at 4 °C for 24 h to allow for subsequent histological analysis.

The sample size (n = 6 per group) was calculated based on the results of a pilot study. A post hoc power analysis using the standard deviation (235 mm^3^) and the tumor volume difference (400 mm^3^) observed in the pilot study indicated that n = 5.41 would be sufficient to achieve 80% power at a significance level of 0.05 (Appendix A). Therefore, a sample size of six animals per group was considered appropriate. This sample size also complies with ethical guidelines aiming to minimize animal use.

### 2.11. Histopathological Analysis

Subcutaneous xenograft tumors were fixed in 4% paraformaldehyde at 4 °C for 12 h and embedded in paraffin. Sections (3 μm thick) were mounted on 3-aminopropyltriethoxysilane (APES)-coated slides. After deparaffinization, the tissue sections were processed using the BOND RX system (Leica, Wetzlar, Germany) for antigen retrieval, followed by immunohistochemical staining. The rabbit monoclonal anti-Ki67 antibody (Nichirei Biosciences Inc., Tokyo, Japan Cat. No. 418071; 1:2), rabbit polyclonal anti-CD31 antibody (Abcam, Cambridge, UK Cat. No. ab28364; 1:50), rabbit monoclonal anti-NF-κB p65 antibody (CST, Danvers, MA, USA, Cat. No. 8242S; 1:40), and biotin-labeled secondary antibodies (Dako/Agilent Technologies, Santa Clara, CA, USA) were applied and incubated at room temperature for 45 min. Specifically, anti-rabbit IgG (Cat. No. K4003; dilution 1:500) and anti-mouse IgG (Cat. No. K4001; 1:1500) antibodies were used. Color development was carried out using the Liquid DAB+ Substrate Chromogen System (Cat. No. K3467) for 10 min at ambient temperature. Nuclear staining was achieved via hematoxylin counterstaining for 30 s. Quantitative analysis followed the protocols described in prior literature [31].

Cell Proliferation (Ki67): The proportion of Ki67-positive cells was determined by calculating the average number of stained nuclei per high-power field (HPF), presented as the mean ± standard deviation (SD). For each treatment group, ten microscopic fields from three independent tumor samples were assessed.

Microvessel Density (MVD): For the evaluation of CD31 expression, each clearly defined CD31-positive region was counted as one vessel. Vessel density was reported as the average vessel count ± SD per HPF based on ten randomly selected fields from three tumor specimens in each group.

NF-κB p65 Activation: Cells exhibiting the nuclear localization of p65 were enumerated, and nuclear staining was interpreted as an indicator of NF-κB pathway activation.

Microscopic Imaging: Stained tissue sections were imaged using a BZ-X710 fluorescence microscope (Keyence Corporation, Osaka, Japan) at 200× magnification.

### 2.12. Statistical Analysis

All experimental data are expressed as the mean ± standard deviation (SD). Multiple group comparisons were performed using one-way analysis of variance (ANOVA), followed by Dunnett’s post hoc test for individual group comparisons. For comparisons between two groups, Student’s *t*-test was used. A *p*-value < 0.05 was considered statistically significant. All statistical analyses were performed using EZR version 1.61 (Saitama Medical Center, Jichi Medical University, Saitama, Japan).

The CalebinA used in this study was provided by Sabinsa Corporation, Tokyo, Japan. A certificate of analysis issued by the company confirmed that the compound had a purity of over 98%, as determined by HPLC analysis. The same amount of CalebinA was used for all experiments, and no batch-to-batch variation was observed.

## 3. Results

### 3.1. IC_50_ Values of Gemcitabine and CalebinA in Pancreatic Cancer Cell Lines

The pancreatic cancer cell lines used in this study were AsPC-1 (Gem-S and Gem-R) and PANC-1 (Gem-S and Gem-R). Each cell line was treated with various concentrations of gemcitabine or CalebinA, followed by a 48 h incubation. To determine the IC_50_ values of gemcitabine and CalebinA, we performed WST-1 assays. The IC_50_ values of gemcitabine after 48 h of treatment were as follows: 0.045 µM for Gem-S AsPC-1, 229.5 µM for Gem-R AsPC-1, 0.061 µM for Gem-S PANC-1, and 59.62 µM for Gem-R PANC-1 (Figure 1a,c). The IC_50_ values of CalebinA after 48 h of treatment were 27.9 µM for Gem-S AsPC-1, 29.8 µM for Gem-R AsPC-1, 22.2 µM for Gem-S PANC-1, and 23.9 µM for Gem-R PANC-1 (Figure 1b,c). Based on these findings, the concentration of CalebinA used in subsequent experiments was set at 10 µM, which is a dose that does not exhibit cytotoxicity.

### 3.2. CalebinA Suppressed NF-κB Activation in Pancreatic Cancer Cells

To evaluate the effect of CalebinA on NF-κB activity, a TransAM NF-κB p65/p50 transcription factor assay was performed. NF-κB activity was found to be enhanced in both pancreatic cancer cell lines as a result of gemcitabine resistance acquired. Upon treatment with CalebinA (10 µM), NF-κB activity was significantly suppressed in all pancreatic cancer cell lines, regardless of the presence or absence of TNF-α. Importantly, CalebinA was found to reduce NF-κB activity even at a low concentration (10 µM), without significantly influencing the proliferation of pancreatic cancer cells. This observation implies that its antitumor potential may be mediated through inhibition of p65 hyperactivation associated with acquired resistance to gemcitabine (Figure 2).

### 3.3. CalebinA Suppressed the Invasive Ability of Pancreatic Cancer Cells

To investigate the effect of CalebinA on the invasive ability of each cell line, a Matrigel invasion assay was performed. Gem-R pancreatic cancer cell lines (AsPC-1 and MIA PaCa-2) exhibited enhanced invasive potential compared to Gem-S cell lines. CalebinA at a concentration of 10 µM markedly reduced the invasiveness of all tested pancreatic cancer cell lines. While the TNF-α stimulation enhanced the invasive capacity of these cells, co-treatment with CalebinA significantly counteracted this pro-invasive effect (Figure 3).

### 3.4. CalebinA Suppressed the Secretion of VEGF and IL-8 from Pancreatic Cancer Cells

The production of angiogenic factors, VEGF and IL-8, by pancreatic cancer cells was quantified using ELISA. Acquired resistance to gemcitabine led to the increased secretion of both proteins. In addition, stimulation with TNF-α further enhanced the release of VEGF and IL-8 in all examined cell lines. Notably, this upregulation was markedly suppressed following treatment with 10 µM CalebinA (Figure 4).

### 3.5. Angiogenesis in Pancreatic Cancer Is Enhanced by Gemcitabine Resistance but Suppressed by CalebinA

The effect of CalebinA on tube formation in human endothelial cells was evaluated. Supernatants from Gem-R pancreatic cancer cells induced significantly greater tube formation compared to those from Gem-S cells. Furthermore, supernatants from pancreatic cancer cells treated with TNF-α significantly promoted tube formation, whereas those treated with CalebinA significantly suppressed it. Furthermore, the increase in capillary-like structure formation induced by TNF-α stimulation was markedly suppressed upon treatment with CalebinA (Figure 5).

### 3.6. CalebinA Suppressed Tumor Growth in a Subcutaneous Xenograft Model

To evaluate the in vivo antitumor effect of CalebinA, a subcutaneous tumor xenograft model was established by injecting AsPC-1 pancreatic cancer cells into nude mice. After tumor formation, the mice in Group I received intraperitoneal injections of 400 µL of a control solution containing the same concentration of DMSO and 5% Tween-80 diluted in Milli-Q water. The mice in Group II received intraperitoneal injections of CalebinA (40 mg/kg in 400 µL of 5% Tween-80 diluted in Milli-Q water) every 3 days. In total, seven injections were administered over a period of 3 weeks. CalebinA significantly suppressed tumor growth in Group II mice, starting from day 18 after the initial administration (Figure 6a,b). No weight loss was observed, and there were no significant differences in body weights between the two groups. Additionally, no atrophy of major organs, including the liver, bilateral kidneys, and spleen, was observed.

### 3.7. CalebinA Suppressed Proliferation and Angiogenesis in Tumor Tissue by Downregulating the Expression of Ki-67, CD31, and NF-κB p65

To investigate the effects of CalebinA on cell proliferation and angiogenesis in tumor tissue, we examined the expressions of Ki-67, a proliferation marker, and CD31, a marker of MVD. CalebinA significantly reduced the expression levels of both Ki-67 and CD31. Furthermore, we assessed the activation of the transcription factor NF-κB p65. CalebinA significantly inhibited NF-κB p65 activation in tumor tissues (Figure 7).

## 4. Discussion

In the present study, we demonstrated that the natural compound CalebinA inhibits tumor progression and angiogenesis in pancreatic cancer cells by suppressing the NF-κB signaling pathway. CalebinA significantly inhibited NF-κB p65 activation in both gemcitabine-sensitive and -resistant pancreatic cancer cell lines, accompanied by decreased secretion of angiogenesis-related cytokines VEGF and IL-8, reduced invasive capacity, and impaired tube formation by endothelial cells. Furthermore, in a subcutaneous tumor xenograft model, CalebinA treatment resulted in suppressed tumor growth and decreased expressions of Ki-67, CD31, and NF-κB p65 in tumor tissues.

While the NF-κB inhibitory effect of CalebinA has been reported in other cancer types [34], our study is the first to report its efficacy in pancreatic cancer, particularly in the context of gemcitabine resistance. Notably, CalebinA effectively suppressed gemcitabine-induced NF-κB activation at a low concentration (10 µM), suggesting its potential to overcome chemoresistance with minimal cytotoxicity.

Previous studies, including our own, have shown that acquired gemcitabine resistance upregulates IL-8 expression [31,32,33], which enhances the angiogenic and invasive capabilities of pancreatic cancer cells. Our findings indicate that CalebinA suppresses this malignant phenotype by inhibiting NF-κB–mediated secretion of IL-8 and VEGF, as well as endothelial tube formation induced by tumor-conditioned media. These results suggest that CalebinA may interrupt the inflammation-angiogenesis cycle within the tumor microenvironment (TME), thereby attenuating tumor progression.

In vivo, CalebinA significantly inhibited tumor growth without inducing notable body weight loss or observable side effects. Since no previous animal studies using CalebinA in pancreatic cancer have been reported, the protocol was determined based on the prior literature on other cancer types [35,36] and adjusted after a pilot study. Immunohistochemical analyses confirmed the reduced expression of Ki-67, CD31, and nuclear NF-κB p65, supporting the hypothesis that CalebinA exerts a multi-level antitumor effect initiated by NF-κB suppression.

CalebinA belongs to a class of curcuminoid-related natural compounds but is structurally a non-curcuminoid derivative [34], suggesting a novel mechanism of action distinct from that of conventional chemotherapeutic agents. Like parthenolide [9] and xanthohumol [31], CalebinA represents a low-toxicity, natural NF-κB inhibitor with promising applications as a therapeutic candidate against gemcitabine-resistant pancreatic cancer.

Finally, we summarize the potential advantages of CalebinA as follows: First, it has low toxicity and natural origins. CalebinA is a naturally derived compound from turmeric and is considered to have high safety with minimal side effects. Second, it demonstrates efficacy against gemcitabine-resistant cell lines. In this study, we demonstrated that CalebinA is effective even in gemcitabine-resistant pancreatic cancer cells, inhibiting tumor growth, angiogenesis, and invasion through suppression of NF-κB activity. This suggests that CalebinA could serve as a novel therapeutic option for cases that are difficult to treat with conventional chemotherapeutic agents. Third, a novel therapeutic strategy was detected targeting NF-κB signaling. Since NF-κB is involved in various cancer-related processes, including inflammation, tumor progression, and chemoresistance, the specific inhibition of this pathway by CalebinA highlights its potential as a molecular-targeted therapy.

However, this study has some limitations. It remains unclear whether CalebinA acts specifically through the NF-κB pathway or affects other signaling cascades such as those involving STAT3 or AP-1. Moreover, the subcutaneous xenograft model does not fully replicate the native pancreatic tumor microenvironment. And data on the pharmacokinetics of CalebinA, including its absorption, distribution, metabolism, and excretion, remain limited, and detailed pharmacological evaluation is necessary for its potential clinical application in humans. In future studies, we plan to conduct evaluations using liver metastasis models. We also plan to perform a comprehensive safety evaluation using a long-term administration model, including assessments of systemic toxicity, hematological changes, and histopathological effects on organs. Finally, we plan to investigate potential additive or synergistic effects in combination with other anticancer agents.

## 5. Conclusions

This study is the first to demonstrate, in both in vitro and in vivo models, that CalebinA effectively suppresses tumor growth and invasion in pancreatic cancer cell lines by inhibiting NF-κB activity and reducing the production of angiogenic factors. To the best of our knowledge, this is the first report to show the efficacy of CalebinA against pancreatic cancer cell lines, indicating its potential as a novel therapeutic agent and a promising breakthrough in pancreatic cancer treatment. These findings suggest that CalebinA may serve as a new therapeutic strategy for overcoming gemcitabine resistance, and further studies are warranted to explore its potential for clinical application.

## Figures and Tables

**Figure 1 nutrients-17-02641-f001:**
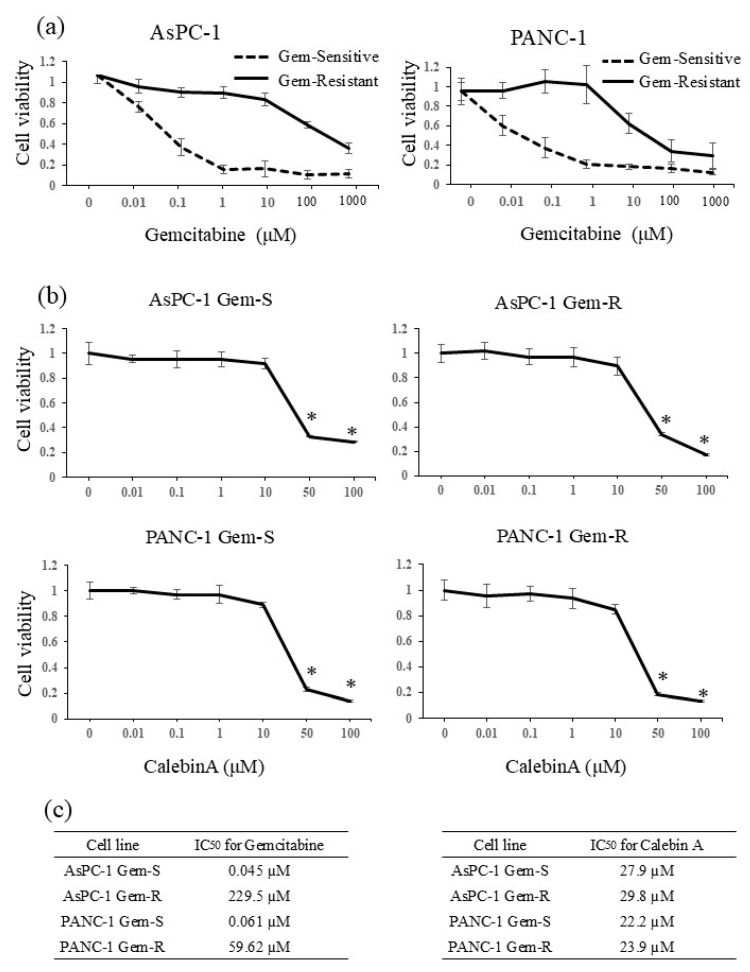
The impact of gemcitabine and CalebinA on the proliferation of gemcitabine-sensitive (Gem-S) and gemcitabine-resistant (Gem-R) pancreatic cancer (PaCa) cell lines. (**a**) Gem-S and Gem-R variants of AsPC-1 (left) and PANC-1 (right) cells were exposed to varying concentrations of gemcitabine for 48 h. Cell proliferation was evaluated using the WST-1 assay. (**b**) Both Gem-S and Gem-R PaCa cells (AsPC-1 and PANC-1) were treated with increasing concentrations of CalebinA for 48 h, and cell viability was determined using the WST-1 assay. (**c**) The half-maximal inhibitory concentration (IC_50_) values of gemcitabine (left) and CalebinA (right) were calculated for each PaCa cell line. Results are presented as the mean ± standard deviation (SD). * *p* < 0.05.

**Figure 2 nutrients-17-02641-f002:**
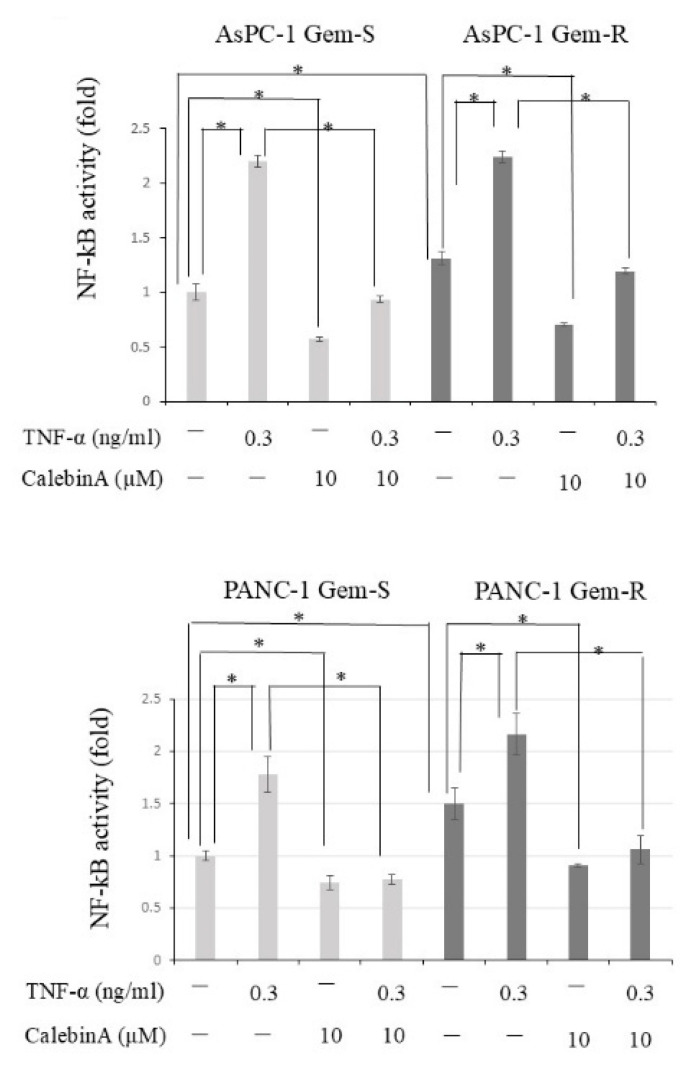
The effect of CalebinA on nuclear NF-κB p65 activity in gemcitabine-sensitive (Gem-S) and gemcitabine-resistant (Gem-R) pancreatic cancer cell lines. NF-κB activity was assessed in nuclear extracts using the TransAM NF-κB p65 assay kit. Gem-S and Gem-R variants of AsPC-1 and PANC-1 cells were treated with CalebinA (10 µM) for 24 h. TNF-α (0.3 ng/mL) was added 15 min prior to the end of incubation to stimulate NF-κB signaling. Data were analyzed via one-way ANOVA followed by Bonferroni’s multiple comparison test. * *p* < 0.05.

**Figure 3 nutrients-17-02641-f003:**
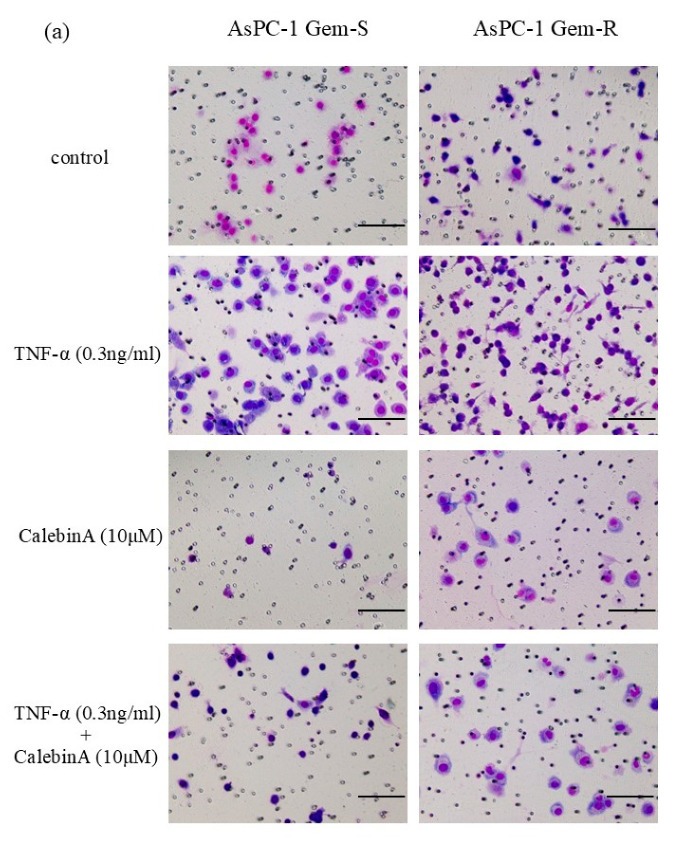
The effect of CalebinA on the invasive potential of gemcitabine-sensitive (Gem-S) and gemcitabine-resistant (Gem-R) pancreatic cancer (PaCa) cell lines, as assessed by Matrigel-coated Transwell invasion assays. PaCa cells (1.0 × 10^5^ cells per well) were seeded into Matrigel-precoated Transwell inserts and treated with TNF-α (0.3 ng/mL) and/or CalebinA (10 µM). Following a 24 h incubation period, migrated cells on the lower membrane surface were stained using the Diff-Quik staining kit and counted. (**a**) Representative microscopic images of Gem-S and Gem-R AsPC-1 cells were subjected to the respective treatments (original magnification: 40×; scale bar = 200 µm). (**b**) Representative images of treated Gem-S and Gem-R PANC-1 cells under the same conditions (magnification: 40×; scale bar = 200 µm). (**c**) The mean numbers of invaded cells (Gem-S and Gem-R AsPC-1, PANC-1) in nine random microscopic fields of view were compared among the treatments using a one-way ANOVA analysis of variance with a post hoc Bonferroni test. * *p* < 0.05.

**Figure 4 nutrients-17-02641-f004:**
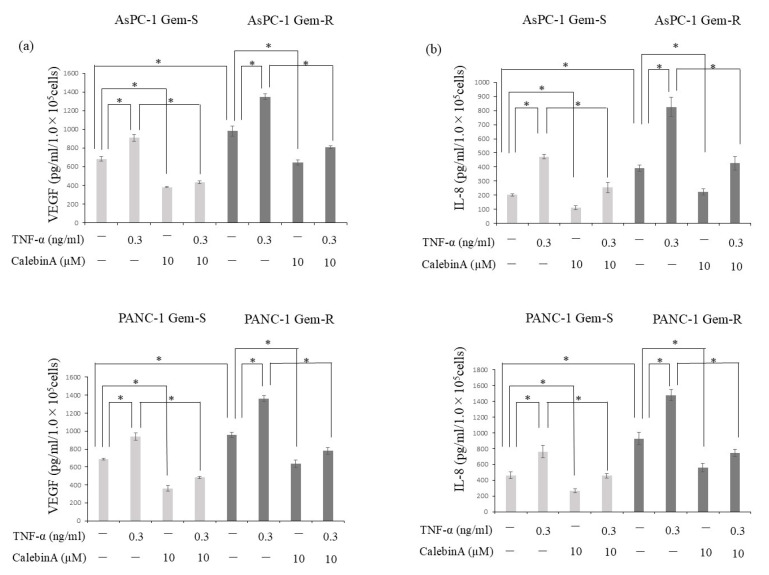
The impact of CalebinA and/or TNF-α on VEGF and IL-8 secretion in gemcitabine-sensitive (Gem-S) and gemcitabine-resistant (Gem-R) pancreatic cancer (PaCa) cell lines. PaCa cells (1 × 10^3^ per well) were seeded in 6-well plates and allowed to adhere overnight. The culture medium was then replaced with a serum-free medium containing either CalebinA (10 µM), TNF-α (0.3 ng/mL), both agents, or neither, and the cells were incubated for an additional 48 h. The levels of VEGF and IL-8 in the supernatants were determined using ELISA. (**a**) Quantification of VEGF secretion in Gem-S and Gem-R AsPC-1 and PANC-1 cells following treatment. (**b**) IL-8 levels were measured under the same treatment conditions. Data are presented as mean ± standard deviation (SD). * *p* < 0.05.

**Figure 5 nutrients-17-02641-f005:**
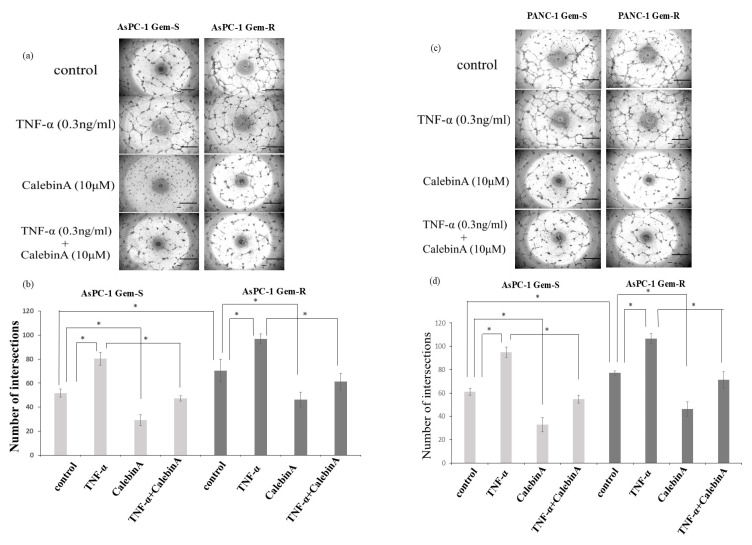
The effect of CalebinA on angiogenic potential in gemcitabine-sensitive (Gem-S) and gemcitabine-resistant (Gem-R) pancreatic cancer (PaCa) cell lines. EA.hy926 endothelial cells (1.2 × 10^4^) were plated onto Matrigel-coated wells and cultured in conditioned media collected from PaCa cells pretreated with TNF-α (0.3 ng/mL), CalebinA (10 µM), or both. (**a**) Representative microscopic images of capillary-like structures formed in response to conditioned media from Gem-S and Gem-R AsPC-1 cells (original magnification: 40×; scale bar = 500 µm). (**b**) Quantification of tube intersections from the corresponding AsPC-1 groups. * *p* < 0.05. (**c**) Representative images of EA.hy926 cells treated with conditioned media from Gem-S and Gem-R PANC-1 cells under the same conditions (magnification: 40×; scale bar = 500 µm). (**d**) Quantitative analysis of intersection counts in the PANC-1 experimental groups. Data are shown as mean ± standard deviation (SD). * *p* < 0.05.

**Figure 6 nutrients-17-02641-f006:**
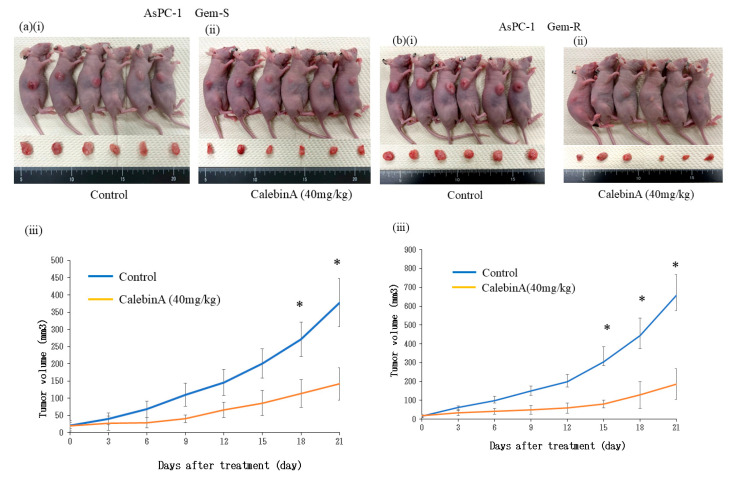
CalebinA suppresses tumor growth in a subcutaneous xenograft model of pancreatic cancer. To establish subcutaneous tumors, 1 × 10^6^ AsPC-1 cells suspended in 100 µL of Hank’s balanced salt solution were inoculated into the flank region. Mice were divided into two groups: an untreated group (Group I) and a treatment group (Group II) that received CalebinA (40 mg/kg, intraperitoneally, every 3 days). (**a**) Images of mice and excised tumors from AsPC-1 Gem-S xenografts. (**b**) Images of mice and excised tumors from AsPC-1 Gem-R xenografts. For both a and b: (**i**) control group, (**ii**) CalebinA-treated group, and (**iii**) tumor volumes in each group were measured every 3 days. Values are expressed as the mean ± SD. * *p* < 0.05.

**Figure 7 nutrients-17-02641-f007:**
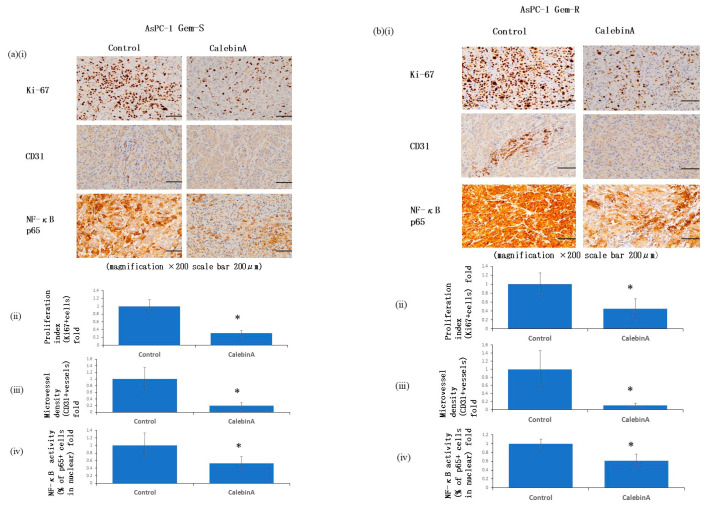
CalebinA suppresses proliferation, angiogenesis, and NF-κB p65 activation in pancreatic cancer xenograft tissues. (**a**) Data from AsPC-1 Gem-S xenografts: (**i**) representative immunohistochemical images (magnification: 200×); (**ii**) quantitative analysis of Ki-67-positive cells; (**iii**) quantification of microvessel density (MVD); (**iv**) and assessment of nuclear p65 activation. (**b**) Data from AsPC-1 Gem-R xenografts: (**i**) representative immunostained sections (magnification: 200×); (**ii**) quantification of Ki-67 labeling index; (**iii**) MVD analysis; and (**iv**) quantification of p65 nuclear translocation as a marker of NF-κB activation. All values are shown as mean ± standard deviation (SD). * *p* < 0.05.

## Data Availability

The datasets generated and analyzed during the current study are available from the corresponding author upon reasonable request.

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
