# Peer review of "The Natural Compound CalebinA Suppresses Gemcitabine Resistance and Tumor Progression by Inhibiting Angiogenesis and Invasion Through NF-κB Signaling in Pancreatic Cancer"

_nutrients, 2025, doi:10.3390/nu17162641_

Round 1
Reviewer 1 Report
Comments and Suggestions for Authors
This study investigates the role of the natural compound Calebin A in pancreatic cancer and finds that it can effectively alleviate gemcitabine (Gem) resistance by inhibiting the NF-κB signaling pathway. These results indicate that Calebin A is expected to become a new candidate drug for the treatment of gemcitabine-resistant pancreatic cancer. The research outcomes have certain theoretical significance and application value, but there are still some issues that need further revision and improvement:
- What is the core molecular mechanism by which Calebin A inhibits gemcitabine-resistant pancreatic cancer? Are the signaling molecules involved in the article sufficient to confirm that its effect is achieved through the NF-κB signaling pathway?
- How were the concentration and dosage of Calebin A used in this study determined? What is the basis?
- Compared with existing pancreatic cancer treatment drugs, what are the main advantages of Calebin A as a potential drug? What other problems remain to be solved?
- The document mentions that gemcitabine-resistant cell lines were established "by continuous passage in medium containing gemcitabine (at a concentration equivalent to its ICâ‚…â‚€) for 3-4 weeks, with repeated concentration adjustments", but it does not specify details such as the specific number of passages, the range of each concentration adjustment, and the specific ICâ‚…â‚€ values of the final resistant cell lines.
- The animal experiment used a subcutaneous xenograft model, while pancreatic cancer has a unique tumor microenvironment (such as the presence of abundant stromal fibrosis), and the subcutaneous model cannot fully simulate its pathological characteristics.
- The document states that Calebin A can inhibit the activity of NF-κB p65, but it is not clear whether it directly binds to NF-κB p65 or exerts its effect through upstream molecules (such as IκB kinase).
- In the animal experiment, Calebin A was administered intraperitoneally every 3 days for 3 weeks, but the basis for determining this administration frequency and cycle was not explained.
- The discussion section mentions that the limitations of the study include "unclear whether it affects other signaling pathways", but it does not mention the evaluation of the long-term safety of Calebin A use.
- The repetition rate of the article is relatively high, and the original images in the supplementary files do not show scale information.
Author Response
Thank you very much for your thorough review.
Please find the attached file for your reference.
We sincerely appreciate your continued support.

Reviewer 2 Report
Comments and Suggestions for Authors
This manuscript presents promising evidence that Calebin A, a turmeric-derived compound, could help treat gemcitabine-resistant pancreatic cancer. Using cell and mouse models, the researchers show that Calebin A suppresses NF-κB activity, reduces invasiveness and angiogenesis of resistant cancer cells, and significantly shrinks tumors in vivo. While questions remain about dosing and clinical translation, the findings highlight Calebin A as a potential new therapy for drug-resistant pancreatic cancer.
The authors used a clear and well-planned set of experiments, including in vitro tests and an in vivo xenograft model, which strongly support their conclusions. Together, these experiments clearly show that Calebin A’s anti-tumor effect is tied to its ability to block NF-κB, leading to reduced angiogenesis and invasiveness. I think that, given the challenge of gemcitabine resistance, the findings are potentially significant for future therapeutic development.
Despite my very positive opinion of the idea and main topic of this manuscript, I do see a few weaknesses, which need to be corrected/answered:
- The manuscript does not include the structure of the studied compound. Please add the chemical structure of Calebin A, at least in the introduction, along with a brief literature review about the compound.
- The study does not compare Calebin A’s effects with other known NF-κB inhibitors or standard second-line therapies, which would provide a stronger sense of relative efficacy.
- The reason for the in vivo dose of Calebin A is only explained by mentioning earlier studies and a “pilot study,” and more explanation would be helpful.
- Please provide details about the number of biological and technical replicates for each major experiment, and indicate if all experiments were performed blinded to group allocation.
- What was the purity of the Calebin A utilized, and were any batch-to-batch differences observed?
- Please add more key words;
- The manuscript fluctuates between “CalebinA” and “Calebin A.” Standardize throughout.
Based on the above I recommend a minor revision, before publication.
Author Response

(The authors gave the same response as above.)

Round 2
Reviewer 1 Report
Comments and Suggestions for Authors
There are no other issues, except that the similarity rate of the paper is still too high.